# Pulmonary Hypertension with Prolonged Patency of the Ductus Arteriosus in Preterm Infants

**DOI:** 10.3390/children7090139

**Published:** 2020-09-16

**Authors:** Ranjit Philip, Vineet Lamba, Ajay Talati, Shyam Sathanandam

**Affiliations:** LeBonheur Children’s Hospital, University of Tennessee, 51 North Dunlap St., Memphis, TN 38104, USA; vlamba1@uthsc.edu (V.L.); atalati@uthsc.edu (A.T.); ssathan2@uthsc.edu (S.S.)

**Keywords:** patent ductus arteriosus, pulmonary hypertension, preterm infant, congestive heart failure, pediatrics, neonatology, surgical ligation of PDA, trans-catheter PDA closure, pulmonary vascular resistance

## Abstract

There continues to be a reluctance to close the patent ductus arteriosus (PDA) in premature infants. The debate on whether the short-term outcomes translate to a difference in long-term benefits remains. This article intends to review the pulmonary vasculature changes that can occur with a chronic hemodynamically significant PDA in a preterm infant. It also explains the rationale and decision-making involved in a diagnostic cardiac catheterization and transcatheter PDA closure in these preterm infants.

## 1. Introduction

In extremely low birth weight (ELBW) infants, the incidence of pulmonary arterial hypertension (PAH) after a month of age ranges 4% to 16% [1]. The presence of severe PAH with bronchopulmonary dysplasia (BPD) in a preterm infant bestows a 40% mortality by 2 years of age [2]. It is difficult to predict the risk of developing PAH in preterm infants as it is multifactorial, with conditions such as BPD, necrotizing enterocolitis (NEC), and sepsis being risk factors [3]. In addition to these factors, there is an enhanced risk of developing PAH early in premature infants with post-tricuspid shunts [4]. The common post-tricuspid shunts include ventricular septal defects (VSD), atrio-ventricular septal defects and patent ductus arteriosus (PDA). The two important factors for outcomes after closure of these defects are the age at the time of closure and the presence of pre-operative pulmonary vascular disease (PVD) [5]. Although there is a general notion that irreversible PAH does not develop in the first year of life, the progression of PAH can be unpredictable in ELBW infants with multiple co-morbidities. The effects on the pulmonary vasculature from increased pulmonary blood flow in a long-standing, large, hemodynamically significant PDA (hsPDA) may be more detrimental than a VSD due to the high-pressure, pulsatile flow throughout the cardiac cycle in a PDA [6]. Management of PAH in preterm infant with a large hsPDA is challenging. This review attempts to highlight the pulmonary vascular changes that can be associated with an hsPDA in a preterm infant. It also introduces diagnostic cardiac catheterization as a tool to assist in decision-making for PDA closure in select infants.

## 2. Pathophysiology

The fetus is in a state of physiologic PAH [7], where the pulmonary vascular resistance (PVR) is elevated and the pulmonary blood flow (PBF) is low. Only 8–20% of the cardiac output goes to the lungs. There are many factors contributing to high fetal PVR. The vascular lumen is narrowed by cuboidal endothelial cells. The lung vasculature is also compressed by the alveoli that is fluid filled. The resting oxygen tension in the arteries and alveoli is low. This leads to hypoxic pulmonary vasoconstriction. The vasoconstriction is also controlled by humoral mediators such as endothelin-1, thromboxane, and leukotrienes and the lack of nitric oxide and prostacyclin (PGI2), which are vasodilators. The normal physiologic transition after birth with respiration involves a precipitous drop in PVR as the ventilation and oxygen tension in the alveolus increases. The removal of the placenta increases the systemic vascular resistance leading to an increase in PBF.

### 2.1. Magnitude of PDA and Prematurity

During the canalicular stage of lung development, the fetal PVR is high. This is due to its immature pulmonary vasculature having a reduced cross-sectional area. The ductus arteriosus in preterm infants is similar in morphology to the fetal ductus and hence classified as a Type-F PDA [8]. Premature infants have a different response to a left-to-right shunt in comparison to term infants.
(i)Faster development of pulmonary overcirculation: The intrinsic tone in their pulmonary arteries is relatively lesser due to a smaller amount of muscle in the media. This precludes appropriate pulmonary constriction and leads to pulmonary overcirculation and cardiac volume overload earlier (2–3 weeks of age) than in term infants (3–6 weeks) [3].(ii)Delayed pulmonary vascular maturation: Similar to respiratory distress syndrome, preterm infants have differential perfusion, i.e, poorly aerated lung segments have decreased PBF and well-aerated segments have good PBF. A left-to-right shunt increases PBF to the already well perfused segments and can lead to pulmonary edema. These pre-capillary vessels are poorly reactive and transmit arterial pressure. A persistent increase in PBF over a prolonged period delays the normal maturation of pulmonary blood vessels. This leads to persistence of smooth muscle with proliferation and hypertrophy and the development of PVD.(iii)Risk of BPD: Preterm infants with a chronic hsPDA are observed to have an increased incidence of BPD [9]. There is equalization of aorto-pulmonary pressures in a large hsPDA. This leads to an increase in the pulmonary venous pressure and the left atrial pressure. The pulmonary capillaries in preterm infants are more permeable. This leads to the leakage of plasma proteins into air-sacs, which affects the function of the surfactant and reduces lung compliance. The mean airway pressures rise to provide adequate oxygenation to non-compliant lungs. This can cause lung damage and possibly lead to BPD [3].

### 2.2. Impact of an hsPDA on Pulmonary Vasculature

After birth, there is a normal physiologic fall in the pulmonary arterial pressure (PAP). There is also thinning of the smooth muscle medial layer. When there is a large PDA, the systemic and pulmonary pressures equalize across the large defect and delay the fall in PAP. Due to the high pressure, the pulmonary arterioles do not mature or thin normally as evidenced by persistence of smooth muscle in the media. This leads to a slower lowering of the PVR in the first 3–4 months of life. Animal pulmonary overcirculation models with an artificially created arterio-venous show increased mean PAP and thickening of the pulmonary artery media similar to that seen in premature infants with PAH who died [10]. There are also animal models of flow-induced PAH in which prolonged, excessive flow has shown not only structural changes but also functional alterations based on changes in vascular reactivity [11]. Although the PVR does not normalize, it falls enough to allow excess PBF. The PBF continues to increase as the PVR decreases. The PAP remains high. This is due to equalization of the pressures and high flow through the large shunt and not due to an elevated PVR.

Most infants with a large, high pressure–high flow shunt usually do not have an elevated PVR. This is easily understood with the formula below:

Pressure = Flow × Resistance; high pressure could be from either high flow from the shunt or high pulmonary vascular resistance. A large, high-flow shunt with low resistance can have high pressure.

### 2.3. Consequence of Prolonged Patency of the PDA on Pulmonary Vasculature

The terminal branches of the pulmonary arteries are called the pulmonary arterioles. The tunica media of the pulmonary arterioles is the first to undergo changes during early infancy. In addition to the abnormal shear stress and circumferential wall stretch to the pulmonary vasculature, there is endothelial cell dysfunction and an imbalance in the vaso-active mediators such as endothelin-1, prostacyclin, and nitric oxide that cause vasoconstriction. There is also increased intracellular matrix deposition and vascular remodeling involving smooth muscle hypertrophy and proliferation due to abnormal expression of fibroblast and vascular endothelial growth factors. There are also inflammatory cytokines such as TNF-alpha, TGF-beta, IL-1, IL-6, and IL-8 that alter the development and severity of BPD in preterm infants and can lead to remodeling of the pulmonary veins [12].

The persistence of an hsPDA beyond a year leads to intimal thickening. If it persists beyond 2 years, it can lead to fibrosis, which will encroach the vessel lumen and decrease pulmonary artery compliance. This increases PVR and reduces PBF. Hence the left-to-right shunting decreases and there is transient improvement in heart failure symptoms. Eventually, right-to-left shunting develops as the pulmonary artery (PA) resistance increases beyond that of the systemic vascular bed. In at least 50% of patients with a chronic large untreated hsPDA, irreversible pulmonary vascular changes may occur by 2 years of age [13]. Even in a patient with PAH and a reversible pulmonary vascular bed, the PVR takes time to normalize after PDA closure [9]. This may, at least in theory, suggest the need for closure of the PDA earlier during the medial muscularization phase.

## 3. Management of a Large PDA with Pulmonary Hypertension

It is clear that the PDA should not be closed when there is right-to-left shunting secondary to supra-systemic PA pressures. The PDA is in fact beneficial as a pop-off in providing cardiac output at the cost of arterial desaturation. It acts as an outlet to the right ventricle to pump into systemic circulation. Usually, the window of opportunity to close the PDA has been lost at this stage. To help decrease the PVR, pulmonary vasodilator therapy may be used in these patients with the intent to possibly reverse remodel the vasculature. If the PVR shows reactivity and decreases, it may potentially allow for PDA closure in the future.

Infants with an elevated PVR and an hsPDA are difficult to treat. The use of supplemental oxygen to treat the PAH may have conflicting effects in the presence of a large hsPDA. The pulmonary vasodilatory effect of oxygen may potentially increase left-to-right shunting and PBF. This may lead to vascular shear stress and damage. The use of pulmonary vasodilator therapy to boost reversal of vascular remodeling may worsen the shunting. Contrarily, closing a PDA with severe PAH runs the risk of a pulmonary hypertensive crisis or right ventricular failure. In such circumstances, further evaluation is essential to assess whether there is reactivity, and/or reversibility of the PVR. A reactive pulmonary vascular bed may indicate potential regression of PAH after PDA closure. Earlier stages of muscular hypertrophy are reversible, but later stages of intimal proliferation and fibrosis are not. Closing hsPDAs by medical therapy or surgical ligation may be suboptimal as these therapies do not lend proper understanding of the PVR, reactivity to vasodilators, and alterations in hemodynamics with PDA closure.

### Role of Cardiac Catheterization

Information from clinical examination, chest X-ray, arterial oxygen saturation, blood gases, and echocardiograms is usually helpful in quantifying the hemodynamic significance of a PDA and mild PAH. Moderate-to-severe PAH is often arduous to quantify accurately by the aforementioned tests to conclusively validate the safety of PDA closure regardless of the method of closure. There are strategies using echocardiography to evaluate the PDA flow in the presence of PAH and a large PDA. If bidirectional shunting goes to all left to right shunting with 100% supplemental O_2_ or iNO (inspired nitric oxide), it suggests reversibility. In some infants, the non-invasive clinical data are equivocal. This is where cardiac catheterization can be of value. It offers the opportunity for meticulous direct hemodynamic assessment including evaluation of vasoreactivity and reversibility of the pulmonary vasculature to vasodilators. Based on these data, the clinician can assess whether a PDA needs to be closed and whether there will be benefit to PDA closure in the presence of PAH. It also provides a therapeutic option for trans-catheter PDA closure (TCPC) in ELBW infants [14,15,16,17,18]. It has the capability to test the real-time hemodynamics after temporarily occluding the PDA, i.e., “test occlusion” thus enabling appraisal of the safety of PDA closure in infants with PAH. This is not possible with medical therapy or surgical ligation. It can also evaluate or confirm pulmonary vein stenosis, which is known to develop in a small percentage of preterm infants with BPD, as this can be challenging to affirm through echocardiography at times. This decision to go ahead with cardiac catheterization has to be weighed in with the risks of transport to the cardiac catheterization laboratory and the risks of the procedure [14].

The Pediatric Pulmonary Hypertension Network (PHN) has outlined a protocol for diagnostic cardiac catheterization for infants with BPD and pulmonary hypertension (PH) [19]. Under baseline conditions, saturation and pressure data are obtained. A Qp:Qs (pulmonary-to-systemic blood flow ratio) is calculated to assess the degree of shunting. Systemic vascular resistance (SVR) and PVR are calculated and indexed (PVRi). A PVRi > 3 Wood Unit × m^2^ is abnormal. A PVR/SVR ratio of >0.5 indicates a high risk for progressive right heart failure or decompensation. To test pulmonary reactivity, these measurements are then repeated with 100% oxygen, 100% oxygen + inspired nitric oxide (iNO) up to 40 ppm (parts per million). Though direct measurements are obtained during a diagnostic cardiac catheterization, there are known limitations in the assessment of cardiac output and PVR. The branch PAs have different saturations, as there is differential PBF from the PDA. Oxygen consumption is estimated in premature infants and, hence, may not be accurate. This exercise of hemodynamic assessment and vasoreactive testing is only done if there is suspicion of high PVR from pre-procedure assessment as it adds to the procedure time.

Test occlusion of the PDA to assess safety for TCPC can be done by two methods. It can either be done by balloon occlusion or by using the anticipated device itself (Figure 1). Balloon occlusion, if not well positioned and stable, has the disadvantage of inadvertently causing aortic or pulmonary obstruction and may make the assessment of hemodynamic data challenging. We prefer device test occlusion. The appropriately sized device is chosen based on the size of the PDA by echocardiography and pulmonary angiography. This procedure is reserved only for cases where the hemodynamics are borderline and the interventionalist wants to confirm that closing the PDA will be well tolerated hemodynamically by the infant. The three important parameters after test occlusion of the PDA to assess safety are as follows:(a)20% drop in the baseline systolic PAP or at least no increase in systolic PAP.(b)No drop in the systemic pressure. This suggests that the PDA was not required for providing cardiac output.(c)No drop in oxygen saturation or clinical signs of a PH crisis [20,21].

If the hemodynamics are favorable, TCPC allows for the aggressive use of pulmonary vasodilator therapy to help with vascular remodeling. The vasodilator therapy also helps to preserve right ventricular function by decreasing right ventricular afterload. Unfortunately, a small percentage of preterm infants who usually have a PVR/SVR ratio >0.5 will have a less than 20% drop in PAP and a small decrease in cardiac output with test occlusion. These infants are more likely to clinically decompensate after TCPC as their PAH does not regress, and they are at risk for progressive right ventricular failure [22]. Therefore, there should be a low threshold to refrain from closing the PDA if the hemodynamic response to test occlusion is borderline, i.e., the PAP falls less than 20% or the systolic blood pressure falls, suggesting that the PDA was needed to augment cardiac output. Selecting infants with reversible PAH is important, since they are the ones who more often benefit from PDA closure. For borderline cases (higher PVR), the use of vasodilator therapy for a couple of weeks prior to PDA closure is thought to have favorable outcomes [15]. In select infants with a high baseline PVR and bidirectional PDA shunting, if 100% oxygen and iNO vasodilate the pulmonary vasculature adequately, leading to pure, left-to-right shunting, increased Qp:Qs ratio, and a drop in PVR, PDA closure may be beneficial to help introduce pulmonary vasodilator therapy. However, the baseline bidirectional shunting may be concerning for potential right ventricular failure with complete closure of the PDA. In these circumstances, based on our previous animal experiment [23], we have preferred to fenestrate the device by cutting out the Gore-Tex membrane off two cells of the device as shown in Figure 2A,B. This provides a pop-off and may prevent right ventricular failure whilst providing partial PDA closure in order for pulmonary vasodilators to be used for vascular remodeling.

Our institutional experience [24,25] with TCPC and hemodynamic testing in ELBW infants (<1000 g at birth, and less than 27-weeks of gestational age) with an hsPDA has showed a difference in hemodynamics based on the time of the procedure from birth. Those who had TCPC at less than 4 weeks of age had a higher Qp:Qs (median 2.5:1) suggestive of a significant shunt, and a lower PVR (1.6 WU·m^2^) in comparison to that of those that had TCPC after 8-weeks of age (median Qp:Qs of 1.8:1, and median PVRi of 3.3 WU·m^2^). Infants with PAH were referred later for TCPC (median procedure age of 84 vs. 32 days). Infants with PAH, though older than those without PAH at the time of TCPC, took longer to extubate after PDA closure. In addition, earlier closure, as expected, allowed for better growth velocity. The mean weight gain between 4 and 8 weeks of age was 25 g/day for the early closure group in comparison to 16 g/day for the group who had TCPC after 8 weeks of age [25]. There is also a misconception that it is technically easier to close the PDA via the transcatheter route in an older infant, which may often be the reason for delayed referral for TCPC. In our experience, pre-procedural risk factors for PDA closure are relatively worse baseline respiratory status and PAH, both of which are seen in ELBW infants referred for TCPC at a later age. There may be benefit in closing hsPDAs earlier, before the development of elevated PVR in ELBW infants. Although our experience continues to be limited by long-term outcomes and controlled trials, there are vital short-term benefits worth considering as mentioned above. One must also be mindful that PAH in preterm infants is multifactorial and that an hsPDA is a contributing factor.

## 4. Conclusions

Persistence of a large hsPDA in ELBW infants could accelerate the development of PAH. Pulmonary vascular changes, even when reversible, may take time to normalize. Early closure within the first 4 weeks of life preceding the development of significantly elevated PVR, may be considered helpful. In select preterm infants with at-least moderate PAH, cardiac catheterization is a good armament to the clinician when the data are equivocal and provides a therapeutic option for patient selection for safe ductal closure. By eliminating the large shunt, aggressive pulmonary vasodilator therapy for PAH can be initiated. A small subset of patients with severe PAH and borderline hemodynamics may not benefit from PDA closure and may eventually develop right heart failure from progressive pulmonary vascular disease. Identifying this small percentage of patients by meticulous vasodilator testing, test occlusion, and partial closure of the PDA is a promising new strategy that is only possible during cardiac catheterization.

## Figures and Tables

**Figure 1 children-07-00139-f001:**
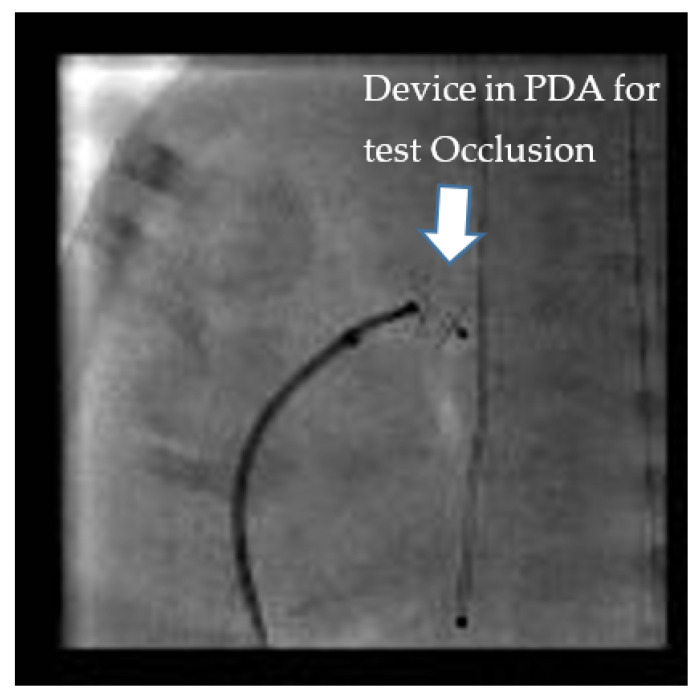
“Test Occlusion”: Testing the hemodynamic effects of temporary occlusion of the patent ductus arteriosus (PDA) during cardiac catheterization. The ideal response is a drop in pulmonary artery pressure by 20% with no drop or an improvement in systemic blood pressure.

**Figure 2 children-07-00139-f002:**
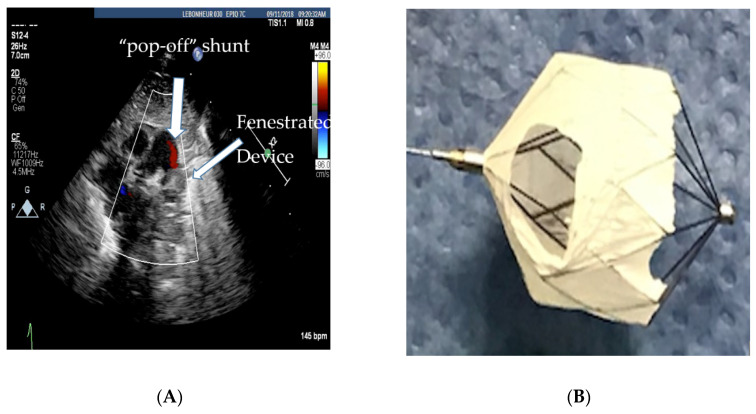
(**A**) Echocardiogram showing the fenestrated MVP-5Q device with the residual “pop-off” shunt, (**B**) fenestrated microvascular plug (MVP-5Q).

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
