# Peer review of "Pulmonary Hypertension with Prolonged Patency of the Ductus Arteriosus in Preterm Infants"

_children, 2020, doi:10.3390/children7090139_

Round 1

Reviewer 1 Report

This manuscript reviews the impact of pulmonary hypertension due to delayed closure of the PDA in the premature infant.  The authors give a nice review of the cardiopulmonary pathophysiology of the newborn and premature infant with a PDA.

A few comments and suggestions for improvement:

Line 87-88:  Suggest discussing the pathophysiology using the most basic of pediatric cardiology formulas: Pressure = Flow x Resistance which may help the learner to understanding more clearly the relationship between pulmonary flow (PDA) and pulmonary resistance (chronic lung disease and BPD of prematurity) and how they interplay to result in pulmonary hypertension.

Line 126- The authors suggest using cardiac catheterization when data is equivocal.  I am not sure that is considered standard of care for the very low birth weight infant.  There are strategies using echo to evaluate the PDA flow in the presence of PAH and large PDA.  If bidirectional shunting goes to all left to right on O2 or INO, there would suggest reversibility. Furthermore, emphasis should be made that if a premature infant with a PDA goes to the cath lab, it would be highly unlikely that the patients leaves without a PDA device.  Decision for a purely diagnostic cath in this scenario needs to be balanced with the risks of transportation and cath lab without a plan for PDA occlusion.  The authors should address this if they propose the diagnostic cath strategy.

Line 133- The authors propose PDA “test occlusion”- this is not so easily accomplished in a premature infant.  In premature infants, the balloon used for test occlusion can obstruct the aortic or PA flow.  Furthermore, it would be difficult to obtain blood saturations for accurate calculation of Qp and Qp/Qs.  If the authors do this routinely, they should review the techniques and discuss potential errors and pitfalls with this technique.

Line 136-9- The authors discuss hemodynamic calculations such as PVR, SVR, PVR/SVR and Qp:Qs.  Again, I would emphasize that there are inherent errors in calculating accurate Qp when there are two sourced of pulmonary flow (PDA).  Even when a device is used to test the occlusion, there may be residual shunting through the device.  

Line 144-45 The authors describe the cath protocol outline by the working group for Pulmonary Hypertension Network. However, I believe the protocol is not applicable or practical for all premature infants as there are weight limitations.  The authors should clarify these limitations.  The authors suggest using the PDA device to test occlude:  How to control for residual shunt through device?  How to obtain mixed venous saturation unless a second catheter is used? Furthermore, it would be quite extreme and expensive to use a PDA device to test occlude and if hemodynamic data (even if accurate) is not favorable, to remove the device and throw it away.  The Piccolo device cost about $9000 each.  The authors need to defend this strategy. Have authors encountered a premature infant who failed test occlusion?

Figure 1 is misleading since the caption talks about test occlusion with a PDA device, but the device has already been released in the image.  If the PDA was not suitable for closure, how does the authors retrieve the device that has already been released?

Line 158- How to identify and control for “post ligation syndrome” with depressed LV function?

Since the current and dominant neonatal position is that long term outcomes of premature infants undergoing PDA occlusion/ligation is similar to those who did not have their PDA closed, it might be beneficial for the readers if the authors can discuss some of the short-term benefits such as improved pulmonary function, improved feeding, improved growth, earlier weaning of ventilator therapy, shorter hospitalization, etc (if data is available).

Overall, this manuscript is a good review. Figure 1 needs to be removed and replaced with an actual case where the patient underwent test occlusion with a device and showing an image with the device that has not been released.

Reviewer 2 Report

The manuscript by Philip and colleagues is a review intended “to remind readers of the pulmonary vascular changes associated with an untreated hsPDA …” (line 32-33). However, since transcatheter closure of the PDA in ELBW neonates is still a novel intervention, the article is less of a reminder and more of a summary and rationalization of the significant experience with this procedure at the authors’ institution, as reflected in multiple publications which they also reference (including one in press – pre-print not yet available online).

In that light, the article does a reasonable job of describing the pathophysiology of progressive pulmonary vascular disease in extremely preterm neonates with a chronic, hemodynamically significant PDA. It also explains the rationale, basic steps, and decision-making involved in diagnostic cardiac catheterization and transcatheter closure of the PDA in these patients. However, the description of the pathophysiologic mechanisms involved is superficial. Whereas the authors describe the pathophysiology of fetal to transitional pulmonary circulation including the role of mediators of pulmonary vascular tone, it would seem equally or even more pertinent to describe the main mechanisms of progressive pulmonary vascular disease in ELBW infants with hsPDA, including mediators of vascular tone, myogenic responses, alveolar hypoxia, inflammation, etc.. Even briefly outlined, such mechanisms would present the reader with a better understanding of the rationale underlying present or potential future interventions.

The authors should note limitations of their experience, such as the lack of long-term outcomes and controlled studies.

Minor issues:

Line 21 should we “certain neonatal diagnoses” (plural)

Reference 4 seems to be incorrect in the context of the text, and it is also a duplicate of reference 11.

Line 120 should have spaces before “reactivity” and “alterations”.

Line 123, should read “echocardiograms are”….

Line 160-161, “a low threshold to not close the PDA if… not ideal…”, requires much effort from the reader to understand clearly – with double negatives and a conditional. Please consider restating more clearly.

Formatting of the references is not consistent: Some use sentence case for the title, others use title case; also, all journal names should be in the abbreviated form.

Reviewer 3 Report

Pulmonary Hypertension with Prolonged Patency of the Ductus Arteriosus in Preterm Infants – A review article by Philip et al. intends to review the consequences of prolonged patency of the PDA on the pulmonary vasculature and the role of cardiac catheterization in preterm infants.

The article’s main objective, as stated in the review introduction, is to remind readers of the pulmonary vascular changes associated with an untreated hsPDA in preterm infants and the role of diagnostic cardiac catheterization to assist in decision making for PDA closure.

The hsPDA could be a contributing factor to development in PH but not the only factor for developing PH. While reading the review, the message comes across as an open ductus usually causes PH, without much evidence. One paper by the author, which is still in revision, has been cited in the last paragraph before the conclusion. Is there evidence to suggest that a VSD in ELBW infants predisposes to PH earlier than expected? If there is evidence, then there could be a stronger argument for an hsPDA to cause PH.

‘The effects on the pulmonary vasculature from increased pulmonary blood flow in a long-standing large, hemodynamically significant PDA (hsPDA) may be more detrimental than a VSD due to the high-pressure, pulsatile flow throughout the cardiac cycle in a PDA’ – The reviewer could not find the evidence in the chapter cited by the authors.

‘Premature infants with hsPDA often develop cardiac failure earlier (2-3weeks of age) than term infants (3-6 weeks)’ – please provide the correct reference. The article cited - Bronchopulmonary dysplasia and pulmonary hypertension: a meta-analysis – does not discuss about cardiac failure.

The management of the PDA with pulmonary HT section provides an explanation about cardiac catheterization and the expected parameters to assess pulmonary vascular reactivity. The authors’ institution has experience in performing cardiac catheterization in ELBW infants. How much does sedation interfere with determining the PVR accurately? Is hsPDA defined based on echocardiogram, or are there clinical parameters taken into consideration?

Lines 182 – 193 – Was the difference limited to PVR and or was there a difference in outcomes such as BPD, NEC, IVH, etc. in early vs. late PDA closure? As the authors mention, PDA is a significant risk for vascular remodeling in an ELBW infant with RDS progressing to BPD. Still, there are multiple factors, including antenatal and postnatal management, that could play a role. The cited paper explaining the institutional experience could help us understand the results better.

Line 195 – the first line of the conclusion is strong without enough evidence to back up. Suggest changing it to ‘could’ accelerate the development of PAH.

Round 2

Reviewer 3 Report

Thank you for your response and edits.